# In Situ Generation of Green Hybrid Nanofibrillar Polymer-Polymer Composites—A Novel Approach to the Triple Shape Memory Polymer Formation

**DOI:** 10.3390/polym13121900

**Published:** 2021-06-08

**Authors:** Ramin Hosseinnezhad, Iurii Vozniak, Fahmi Zaïri

**Affiliations:** 1Centre of Molecular and Macromolecular Studies Polish Academy of Sciences, 90-363 Lodz, Poland; ramin.h@cbmm.lodz.pl; 2Univ. Lille, IMT Lille Douai, Univ. Artois, JUNIA, ULR 4515-LGCgE, Laboratoire de Génie Civil et géo-Environnement, F-59000 Lille, France; fahmi.zairi@polytech-lille.fr

**Keywords:** biocomposites, smart material, interface/interphase, extrusion

## Abstract

The paper discusses the possibility of using in situ generated hybrid polymer-polymer nanocomposites as polymeric materials with triple shape memory, which, unlike conventional polymer blends with triple shape memory, are characterized by fully separated phase transition temperatures and strongest bonding between the polymer blends phase interfaces which are critical to the shape fixing and recovery. This was demonstrated using the three-component system polylactide/polybutylene adipateterephthalate/cellulose nanofibers (PLA/PBAT/CNFs). The role of in situ generated PBAT nanofibers and CNFs in the formation of efficient physical crosslinks at PLA-PBAT, PLA-CNF and PBAT-CNF interfaces and the effect of CNFs on the PBAT fibrillation and crystallization processes were elucidated. The in situ generated composites showed drastically higher values of strain recovery ratios, strain fixity ratios, faster recovery rate and better mechanical properties compared to the blend.

## 1. Introduction

The use of biodegradable polymers in recent years has been increasing in a number of industrial applications to prevent the accumulation of plastic waste. Biodegradable polymers also show a wide range of relevant properties such as processability, versatility and biocompatibility, among others, characteristics that are responsible for the increased interest in the shape memory field [1,2,3,4]. Shape memory polymers (SMPs) could display dual [5] or multi shape-changing behavior [6,7,8] from a “dormant” temporary shape to permanent shape upon the application of an external stimulus like a change in temperature, exposure to light, moisture, solvent and so on. In general, SMPs combine chemical or physical crosslinks (net-points), which define the permanent domain, and the switching (reversible) domains with right proportion, which are responsible for shape transitions. The SMPs articles can be easily processed into complex forms by conventional methods, such as injection molding, film casting, fiber spinning, profile extrusion, and foaming. However, one main drawback of pure SMPs is their low strength and stiffness compared to shape memory metal alloys. Moreover, they exhibit relatively low recovery stress which is a limiting factor in many applications especially in cases where SMP articles should overcome a large resisting stress during shape recovery. 

Overcoming this problem can be achieved by mixing SMPs with various high strength fillers (e.g., carbon nanotubes [9,10], nanoparticles [11,12] and discontinuous or continuous fibers [13,14,15]). However, it has been observed that there is a compromise between strength enhancement and recovery degree (strain recovery ratio) due to the effects of filler size, much higher stiffness of fillers or potential infringement of the chemical or physical crosslinks especially at high filler loadings as well as poor interactions between the fillers and the matrix and improper morphology of the fillers. To minimize these negative effects, nanosized fillers with low loading content are commonly used. 

An alternative way is the use of in situ generated microfibrillar or nanofibrillar polymer-polymer composites (MFCs or NFCs, respectively) created by fibrillizing the dispersed component of an immiscible polymer blend. The MFC process has been shown to be successful for biodegradable polymer pairs, including PLA/PBS [16], TPU/PLA [17], thermoplastic starch/PLA [18], PLA/PGA [19], PLA/PA [20,21], PLA/PHA [22] and PCL/PHA [23], with properties such as modulus and strength increasing manifold when compared with the neat matrix polymer. Compared to solid particle fillers or ready-made polymer fibers, in situ generated polymer fibers demonstrate an excellent level of dispersion. The in situ fibrillated polymer is also characterized by higher specific interfacial area than particle dispersed fillers. More interestingly, the formation of MFCs or NFCs naturally leads to a cascade of three or more elastic modulus plateaus (depending on the number of dispersed phases) of decreasing magnitude with increasing temperature, i.e., the ability to create thermally activated multi-SMPs. The resulting multi-SMPs display at least two well-separated transitions, the glass/melting transition of the matrix and the melting of the fibers, which are subsequently used for the fixing/recovery of temporary shapes. Moreover, the large interfacial area intrinsic with in situ generated fiber/matrix morphology greatly enhances interfacial interactions and may lead to better shape fixing and recovery. It should be noted that at present, most of the created multi-SMPs based on polymer blends or multiphase polymers [24,25,26,27,28] are characterized by a broad interval of phase transitions, which adversely affects the degree of fixation of temporary shapes, or have fully separated phase transition temperatures, but suffer from insufficient strong bonding between the polymer blends phase interfaces which affects their low strain recovery ratios. Moreover, the preparation of multiphase polymers often involves completely different polymerization technologies and tedious chemical processes. 

The in situ fibrillation method is cost-effective and potentially suitable for large-scale production; it offers a much greater degree of design flexibility, since the two functional components (matrix and fibers) can be separately tuned to achieve optimal control of properties. Despite this enormous potential, however, there is only one notable article on shape memory elastomers reinforced by in situ PGA fibrillation, published by Wang et al. [29]. The efficiency of in situ fibrillation of the dispersed polymer phase depends on viscosity and elasticity ratios [20] and can also be significantly increased due to the implementation of shear-induced crystallization [30]. In this work, the effect of in situ poly(butylene adipate-co-terephthalate) (PBAT) fibrillation on the shape-memory properties of PBAT/PLA blends in a 90:10 ratio was investigated in terms of morphology, thermal and mechanical properties in order to correlate it with the shape memory performance of the nanocomposites. Furthermore, the influence of the addition of cellulose nanofillers (CNFs) on the in situ fibrillation and crystallization processes of the dispersed polymer phase as well as on the shape memory properties (strain recovery ratio and strain fixity ratio) of the in situ generated nanocomposites was investigated. The versatility of this new strategy lies in the possibility to vary the number and combination of immiscible polymer components, thereby opening the way for the development of triple or multiple (with more than two dispersed polymer phases) SMPs.

## 2. Experimental

### 2.1. Materials

Commercial grade of PLA 4060D, supplied by NatureWorks LLC (Minnetonka, MN, USA), with a density of 1.24 g cm^−3^, M_w_ of 120,000 g mol^−1^, and 18 mol.% of d-Lactide content was used as the matrix. PBAT with the trade name Ecoflex type F Blend C1200 by BASF AG (Ludwigshafen am Rhein, Germany) was purchased and used in order to reinforce the PLA. Cellulose nanofibers (CNFs) with diameters between 10 and 20 nm and lengths of 2 to 3 µm, obtained from Nanografi (Ankara, Turkey), were employed to enhance the morphological evolution of PBAT droplets into fibers.

### 2.2. Sample Preparation

Blends and in situ generated nanocomposites of PLA/PBAT were prepared following the procedure described earlier [31]. In situ generated nanocomposites were prepared by shearing of blends to allow formation of nanofibers and their shear induced crystallization. To avoid the degradation due to hydrolysis, PLA and PBAT were dried at 65 °C under vacuum before extrusion. Melt blending was carried out in a co-rotating twin screw extruder (2 × 20/40D EHP Zamak Mercator, Skawina Poland, with a ratio of screw length to its diameter (L/D) of 40). Further blending was followed by extrusion of tapes using single screw extruder (PlastiCorder PLV 151, Brabender, L/D = 25, Duisberg Germany) equipped with the 12 mm wide, 0.8 mm thick and 100 mm long slit die. The extrudates were cast on a transport belt at 25 °C. The extrudates in the form of tapes approximately 0.5 mm thick and 10 mm wide were obtained. The PBAT concentration was 10 wt.%. For the preparation of reinforced composites, 3 and 7 wt.% of CNFs were mixed with PLA and PBAT in the course of the twin-screw extruder where the temperature zones were set increasingly from 135 °C to 160 °C. In situ conversion of blends was performed within a single-screw extruder with a temperature gradient descended from 180 °C (feed section) to 115 °C (slit die). In the whole text, C and B stand for the composite and blend, respectively. The numbers in parentheses indicate the concentration of CNFs.

### 2.3. Mechanical and Thermal Properties

Tensile properties of samples were measured in Instron-5582 (Universal Testing Machine, High Wycombe, UK) at a strain rate of 5% min^−1^. Specimens of the gauge length of 25 mm and the width of 5 mm (ISO 527-2, type 1BA) were cut out from extruded tapes using a steel template. At least seven specimens were tested for each sample at room temperature (T_d_ = 22 °C). The thermomechanical properties of rectangular films (dimensions of 24 × 10 × 0.75 mm^3^) were characterized using a Q800 DMA (TA Instruments, New Castle, DE, USA) at a heating rate of 2 °C min^−1^. The DMA studies were performed in a multi-frequency-strain mode using single cantilever fixture with a clamp size of 17.5 mm, strain set to 0.02 %. Thermal behavior of samples was probed with DSC Q20 differential scanning calorimeter (TA Instruments) during heating from 0 to 150 °C with the rate of 10 °C min^−1^. Samples of the 7–8 mg mass were cut out from blends and composites and crimped in standard Al pans. The DSC cell was purged with dry nitrogen during the measurements (20 mL/min).

### 2.4. Scanning Electronic Microscopy (SEM)

The morphology of blends and in situ generated composites, cryogenically fractured along the extrusion direction and coated with gold, was investigated with JEOL JSM-5500 LV scanning electron microscope (Tokyo, Japan). 

### 2.5. Rheological Measurements

Rheological behavior of the materials was examined using a strain-controlled rotational rheometer (ARES LS2, TA Instruments). Uniaxial extension tests of molten samples were performed using extensional viscosity fixture (EVF, TA Instruments) attached to the ARES rheometer. The 18 × 10 × 0.7 mm^3^ rectangular specimens were uniaxially extended at 70 and 120 °C with a constant Hencky strain rate. The tensile stress growth coefficient as a function of time at a given Hencky strain rate was measured. The tests for each material were repeated at least eight times, and the results were averaged.

### 2.6. Shape Memory Properties

Thermally activated shape memory characterization of samples was conducted using the same Q800 DMA instrument with the film tension clamp under controlled strain and controlled force modes. For dual memory shape studies, a strain deformation of 10% was applied at 120 °C. The strain was held constant till the sample was cooled down to the room temperature, then released. The sample was then heated back up to 65 or 120 °C again, while the length recovery was monitored. In the case of triple memory shape studies, a primary strain deformation of 10% was applied at 120 °C. The strain was then held constant while the sample was cooled down to 65 °C, where second shape deformation was applied to increase the strain from 10 to 25%. The strain was held constant until the sample was cooled down to room temperature, and was then released. The sample was heated back up to 65 °C to recover the shape in first step for 30 min. Later, the temperature was increased to 120 °C. Appendix A (in the Appendix A) represents the method Log for dual-SME and triple-SME, containing the whole program set for shape memory studies.

To get a quantitative estimation of the thermally activated shape memory properties of the materials, the strain fixity ratio *R_f_* and the strain recovery ratio *R_r_* were calculated. In particular, *R_f_*, the ability to fix the temporary shape, is the ratio of the fixed strain to the total strain, as presented by the following equation: $R_{f}=\left(\frac{e_{un}}{e_{max}}\right)\times 100\%.$ *R_r_,* the ability to recover the initial shape, was taken as the ratio of the recovered strain to the total strain, as given by the following equation: $R_{r}=\left(\frac{e_{un}-e_{fin}}{e_{max}}\right)\times 100\%$, where *e_un_* is the strain after cooling and unloading, *e_max_* is the strain obtained before the constant loading was released, and *e_fin_* is the strain obtained after heating in the step of recovery.

## 3. Results and Discussion

### 3.1. Morphology

The morphology of PLA/PBAT blends and in situ generated composites reinforced with CNFs was investigated by SEM analysis. The SEM images of the cryo-fractured surface of PLA/PBAT blends and composites reinforced with 3 and 7 wt.% CNFs are shown in Figure 1. It can be seen that the blends are characterized by a matrix-droplet morphology (Figure 1a,b), while in situ generated composites have a fibril-matrix morphology (Figure 1c,d). As the concentration of CNFs increases, the average size of PBAT droplet inclusions decreases. In particular, the average particle size is 870 and 400 nm for the blends containing 3 and 7 wt.% CNFs, respectively. It can be assumed that the presence of CNFs prevents the aggregation of PBAT particles in the case of blends, thereby increasing the interfacial interaction between PBAT and PLA. Most PBAT nanofibers are in the range of 50–200 and 40–120 nm at 3 and 7 wt.% CNFs, respectively. The PBAT nanofibrils also appear to exhibit very high aspect ratios. However, due to the high entanglement of the PBAT nanofibrils, it was not possible to measure their aspect ratios. In our previous work [31], it was shown that the in situ formation of the PBAT nanofibrils takes place in the region between the extruder screw and the extruder walls followed by their immediate solidification by shear induced crystallization. The sizes of in situ generated PBAT nanofibrils were 50–300 nm [31]. Comparative analysis of PBAT nanofibrils sizes shows that the elongated shape of the CNFs improves the efficiency of the droplet to fiber transition by further contributing to fiber elongation, thus causing the formation of thinner PBAT nanofibers. It should be noted that the absence of visible aggregates of CNFs indicates their relatively good dispersion.

In the case of immiscible polymers, the formation of a physically entangled network can occur at the interfaces. To verify this assumption, the viscoelastic behavior of the PLA/PBAT/CNFs blend and the in situ generated composites was investigated. Figure 2a shows a time-dependence of the tensile stress growth coefficient $\eta_{E}^{+}$ (t,$\overset{˙}{\varepsilon }$) for PLA/PBAT and PLA/PBAT/CNFs blend and composite at T = 70 °C recorded during uniaxial tension at Hencky strain rate $\overset{˙}{\varepsilon }$ = 1 s^−1^. For PLA/PBAT blend, the monotonic stress growth coefficient, $\eta_{E}^{+}$ (t,$\overset{˙}{\varepsilon }$), shows a typical poor performance during the elongation flow, i.e., predominantly plastic flow. The tensile stress growth coefficient reaches the end of the plateau when the chains are strongly stretched. At this plateau and beyond, the chains could start to slide past each other at the entanglement points, which leads to a yielding of the entanglement network and the occurrence of a strain softening effect. The addition of 3 and 7 wt.% of CNF to the blend increases the tensile stress growth coefficient $\eta_{E}^{+}$ (t,$\overset{˙}{\varepsilon }$) as shown in Figure 2a. 

This is attributed to the increased elasticity and new load-bearing entanglements, resulted from the presence of CNFs. In contrast to the blends, in situ generated PLA/PBAT composite containing PBAT nanofibers and the same CNFs content shows more improved performance during uniaxial extensional deformation. The PBAT nanofibers form an effective network of physical entanglements at the PLA-PBAT interface that deforms during uniaxial extension and cause a stronger tendency for upward deviation of $\eta_{E}^{+}$ (t,$\overset{˙}{\varepsilon }$), known as strain hardening. Strong strain hardening effects and increase in the elongational viscosity is observed in the case of the composite with 7 wt.% of CNFs, which suggested the greatest number of physical entanglements. Figure 2b depicts the time-dependence of $\eta_{E}^{+}$ (t,$\overset{˙}{\varepsilon }$) for PLA/PBAT and PLA/PBAT/CNFs blend and composite at T = 120 °C and Hencky strain rate $\overset{˙}{\varepsilon }$ = 1 s^−1^. CNFs also form a network of physical entanglements with PBAT and PLA chains which is indicated by the hinderance of strain softening at the melting point of PBAT for the composite with 3 wt.% of CNFs and a high growth rate of $\eta_{E}^{+}$ (t,$\overset{˙}{\varepsilon }$) in the case of composite with 7 wt.% of CNFs. The formation of a more effective network of physical entanglements at PLA/PBAT, PBAT/CNFs and PLA/CNFs interfaces in the in situ generated composite compared to the blend is also confirmed by the DMA data shown below. 

### 3.2. Thermal and Dynamic Mechanical Analysis

The DMA curves of PLA/PBAT as well as PLA/PBAT/CNFs blend and composite are presented in Figure 3. It is seen that the addition of CNFs leads to an increase in the storage modulus E^/^ values for both blends and composites both in the range below T_g_,_PBAT_ and above this temperature up to T_g_,_PLA_ (Figure 3a). However, the main contribution to the change in the E^/^ values is made by the transition of the PBAT phase morphology from droplets to fibrils. Besides, the absolute values of E^/^ for the composite with 7 wt.% CNFs in the T_g_,_PLA_ region remain higher than those for the blend with 7 wt.% CNFs at temperatures below T_g_,_PBAT_.

Since it is known that even in the case of immiscible polymers, an interaction is possible at the phase interface; the tan δ curves were analyzed. As shown in Figure 3b,c, both blends and composites showed only one tan δ peak in the temperature range of 50–80 °C and the values at tan δ, and thus, the internal friction changed with changes in the PBAT phase morphology. This may be attributed to the physical crosslinking of PLA and PBAT molecular chains at two-phase interfaces. The PBAT phase in the form of nanofibrils, compared to nanodroplets, forms a network with a larger number of physical entanglements with PLA. This can limit the viscous flow of PLA molecular chains more effectively. The values of the maximum tan δ also decrease with increasing CNFs concentration, indicating the formation of physical entanglements at the PLA—CNFs interfaces. 

Similar behavior with variation of CNFs concentration and PBAT morphology was observed for tan δ in the temperature range of −40–0 °C, which characterizes the PBAT phase (Figure 3d). This indicated that the physical crosslinking at the PLA-PBAT and PBAT-CNFs interfaces was also stable and can effectively limit the viscous flow of the PBAT molecular chains. A comparative analysis of the DMA curves shows that more effective physical crosslinking networks are formed at the interphases of a composite than at a blend, which suggests higher SME parameters, such as strain recovery and strain-fixation ratio. A critical role of physical crosslinking at the phase interfaces in shape recovery has also been reported in other studies [32,33]. 

The introduction of CNFs leads to an increase in the glass transition temperatures of PLA and PBAT. The greatest effect is achieved at 7 wt.% CNFs and is 2–4 and 8–10 °C for T_g,PLA_ and T_g,PBAT_, respectively. Since PLA is a reversible phase, knowledge of its exact glass transition temperature in the formed blends and composites is required when programming the SME. In addition, information on the melting and crystallization temperatures of PBAT and information on the amount of crystalline PBAT phase, which plays the role of both reversible and fixed domains, is required for programming the SME. Figure 4 shows DSC curves for the PLA/PBAT/CNFs blends and composites after heating and subsequent cooling cycles. It can be seen that the addition of CNFs does not lead to a significant change in the melting temperature and degree of crystallinity of PBAT (Figure 4a). At the same time, the presence of CNFs contributes to a significant increase in the crystallization temperature of PBAT from 53 °C to 85–87 °C, depending on the content of CNFs (Figure 4b).

Since the glass transition of PLA takes place at 62–65 °C, it was very important for the programming of triple shape memory effect (triple-SME) that the PBAT crystallization process is completed prior to the glass-to-rubber transition of PLA. Thus, the second temporal shape (deformed PLA network) will be programmed at temperatures where the crystalline phase of PBAT, which is responsible for the formation of the temporary PBAT network, is fully crystallized. Otherwise, the programming of the second temporal shape was accompanied by a partial recovering of the first temporal shape. It should also be noted that maintaining the amount of PBAT crystalline phase in blends and composites while adding CNFs helped to maintain the required amount of switching domains for the first temporal shape.

### 3.3. Thermoresponsive Shape Memory Performance

The SME tests were conducted in three stages. In the first stage, a dual shape memory effect (dual-SME) at the temperature T_g,PLA_ was programmed to determine the characteristics of the processes for fixing the temporary and recovering the original shape for the blends and composites (Figure 5). In the second stage, the dual-SME was programmed at the temperature T_m,PBAT_ (Figure 6). In the third stage, the possibility of programming the triple-SME was investigated (Figure 7). It is revealed that the shape recovery of blends, in the case of dual-SME at the temperature T_g,PLA_, is incomplete with R_r_ of 29 and 32% for 3wt.% and 7wt.% CNFs, respectively. At the same time, significantly higher R_r_ values are achieved for composites, namely, 42% for a composite containing 3wt.% CNFs and 85% for a composite with 7wt.% CNFs.

The differences are related to the characteristics of the morphology of the minor PBAT phase and the content of the CNFs. In the case of droplet matrix morphology, a weak network of physical entanglements is likely formed at the interfaces of PLA-PBAT and PLA-CNFs. As a result, PLA chains slip during the formation of a temporary shape and are not able to effectively recover their original shape during subsequent heating. An increase in CNF concentration leads to more entanglements at the PLA-CNF interface but does not change the number of physical entanglements at the PLA-PBAT interface. This leads only to a partial improvement in SME properties. In composite, the PBAT nanofibers form an effective network of physical entanglements at the PLA-PBAT interface, which leads to the achievement of higher R_r_ values. The presence of CNFs causes the formation of longer and thinner PBAT nanofibers, which are capable of forming a more developed network of physical entanglements, which significantly increases the recovery rate, especially in a composite with 7 wt.% CNFs.

Dual-SME at the temperature T_m,PBAT_ has its own features. No shape recovery is observed in blends and composites with a low content of CNFs (3 wt.%) (shown in Supplementary Material), which is probably due to the low density of physical entanglements at the PBAT-CNFs interface. Increasing the CNF content to 7wt.% leads to the formation of effective physical entanglements between PBAT and CNFs and to an improvement in SME properties. The change in PBAT morphology from droplet to fibrillar leads to higher R_r_ values (Figure 6). It should be noted that in contrast to the dual-SME at temperature T_g, PLA_, in the case of the dual-SME at temperature T_m, PBAT_, the change in PBAT morphology also leads to a significant difference in the time of shape recovery *t_r_*. In the case of the PLA/PBAT/7 wt.% CNFs blend, complete shape recovery is achieved within 15 min, while in the case of PLA/PBAT/7 wt.% CNF—4 min.

Based on the above data for dual-SME at the temperatures T_g,PLA_ and T_m,PBAT_, triple-SME was programmed for a blend and composite with 7 wt.% CNFs (Figure 7). It can be seen that both the blend and the composite show good fixation of the temporary shapes and their subsequent recovery to the original shape. However, the in situ generated composite showed a significantly higher thermoresponsive SME than the blend during the shape memory cycle.

### 3.4. Thermoresponsive SME Mechanism of PLA/PBAT/CNF Systems

Based on the above analysis, the thermoresponsive SME mechanism of PLA/PBAT/CNFs blend and in situ generated composite was proposed. In these systems the PLA phase is the first reversible phase, the crystalline PBAT phase is the second reversible phase, while PBAT below T_m_ and CNFs are the fixed phases to maintain the temporary PLA phase and CNFs to maintain the temporary PBAT phase. In fact, physical crosslinking at the PLA-PBAT, PLA-CNF, and PBAT-CNF interfaces maintains temporary phases by restricting the irreversible sliding of PLA and PBAT chains during the shape memory cycle. Accordingly, the first deformation temperature (T_d1_) should be higher than T_m,PBAT_ and the first fixation temperature (T_f1_) should be between T_g,PLA_ and T_c,PBAT_, while the second deformation temperature (T_d2_) should be the glass transition temperature of PLA. The second fixation temperature (T_f2_) should be the temperature below T_g,PLA_ (e.g., room temperature). In SME programming, PBAT crystals are first melted at T_d1_ and the temporary network is formed by the viscous flow of the PBAT molecular chains under external stresses. When the temperature has cooled down to T_f1_, the first temporary shape is fixed by rapid crystallization of the reversible phase PBAT even without external stress. The presence of CNFs causes the crystallization of PBAT phase at a temperature higher than the T_g_ of PLA. The first temporary shape is maintained by the permanent network of CNFs. Then, the temperature is lowered to T_g,PLA_ so that a temporary PLA network can be formed. At the same time, PBAT and CNFs act as physical nodes for this network. During the subsequent heating, first the PLA and then the PBAT chains regain mobility and release the stress stored in the temporary networks, thus restoring the original shape. It should be noted that the presence of CNFs as well as the morphology of the minor polymer phase determine the density of the physical entanglements and thus influence the properties of the SME.

### 3.5. Mechanical Properties

As polymer materials with SME must have high mechanical properties from a practical point of view, this aspect was also investigated. It is known that the creation of a network of flexible polymer nanofibers in a brittle polymer matrix can lead to a simultaneous increase in their strength and plasticity [31]. Table 1 shows the values of modulus of elasticity *E*, yield stress σ*_y_*, tensile strength σ*_b_* and strain at break ε_b_ for PLA/PBAT and for PLA/PBAT/CNFs blends and composites. It can be seen that in the case of composite, a simultaneous increase in strength properties and strain at break can be achieved, whereas in the case of a blend, an increase in plasticity is accompanied by a decrease in strength. The introduction of CNFs leads to an increase in modulus of elasticity, yield stress and tensile strength in both blends and composites. However, unlike PLA/PBAT/CNFs blends, which are highly embrittled, PLA/PBAT/CNFs composites retain a sufficiently high degree of plasticity. Together with the high SME properties, this allows in situ generated composites to be considered as effective polymer materials with triple shape memory.

## 4. Conclusions

The concept of in situ generation of polymer-polymer nanocomposites was applied to the production of triple SMPs. A green hybrid blend PLA/PBAT/CNFs was chosen as the initial polymer system. The in situ fibrillation of the dispersed PBAT component led to the formation of more effective physical entanglements at the PLA-PBAT and PBAT-CNF interfaces due to their higher specific interfacial area compared to droplet dispersed fillers, playing the role of physical cross-links (net-points). The introduction of CNFs promoted more efficient conversion of the PBAT phase from droplets to fibers and, as a consequence, the formation of thinner and longer PBAT nanofibers, which caused stronger interaction at the PLA-PBAT and PBAT-CNF interfaces. At a CNF concentration of 7 wt.%, physical entanglements were formed at the PBAT-CNF interface, which allowed the shape memory effect to be realized at T_m,PBAT_. Moreover, the presence of CNFs caused a significant increase in the crystallization temperature of PBAT by 30 °C, which allowed to separate the processes of programming temporary shapes associated with the crystallization of PBAT and the glass-rubber transition of PLA. The advantages of the composites compared to blends, such as increased strength and stiffness of the polymer inclusions; the presence of physical entanglements at the interfaces and improved interfacial compatibility resulting from a larger specific contact area between the phases contributed to the formation of a better shape memory performance (strain recovery, strain fixity rates, recovery rate) and a better complex of mechanical properties (modulus, yield stress, strain at break). In addition, better shape memory properties of in situ produced composites were achieved without deterioration of the mechanical properties. It should be noted that the inherent versatility of this concept allows an unprecedented degree of design flexibility for functional triple- or even multi-shape memory polymers and systems.

## Figures and Tables

**Figure 1 polymers-13-01900-f001:**
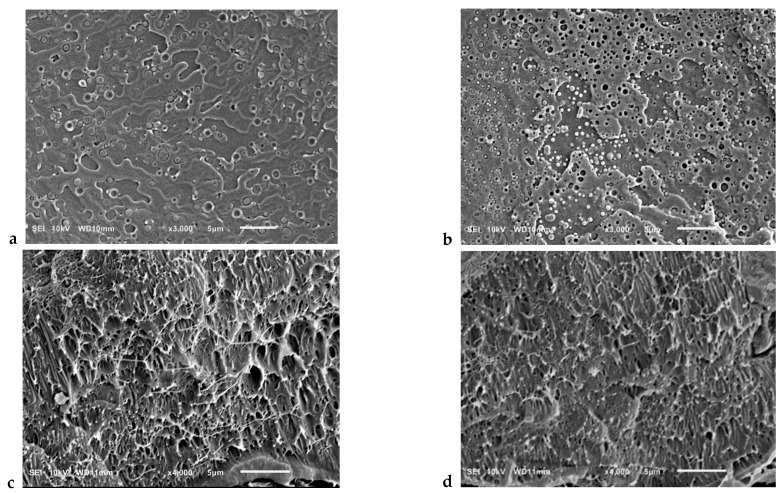
SEM images of cryofracture surfaces of PLA/PBAT/CNFs blends (**a**,**b**) and in situ generated composites (**c**,**d**). (**a**,**c**)—3wt.%CNFs, (**b**,**d**)—7wt.%CNFs.

**Figure 2 polymers-13-01900-f002:**
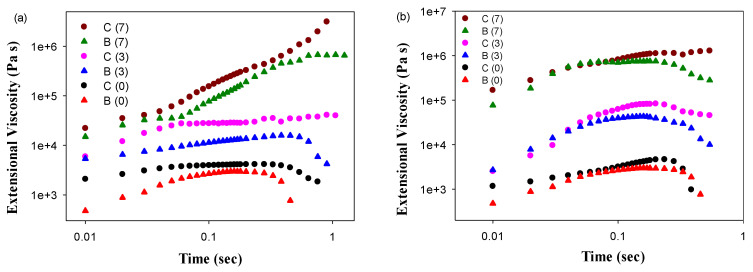
Time-dependence of tensile stress growth coefficient $\eta_{E}^{+}$ (t,$\overset{˙}{\varepsilon }$) for molten PLA/PBAT as well as PLA/PBAT/CNFs blends and in situ generated composites. The shear tests were performed at temperatures 70 °C (**a**) and 120 °C (**b**). Curves were shifted vertically for better visualization. C and B stand for the composite and blend, respectively. The numbers in parentheses indicate the concentration of CNFs.

**Figure 3 polymers-13-01900-f003:**
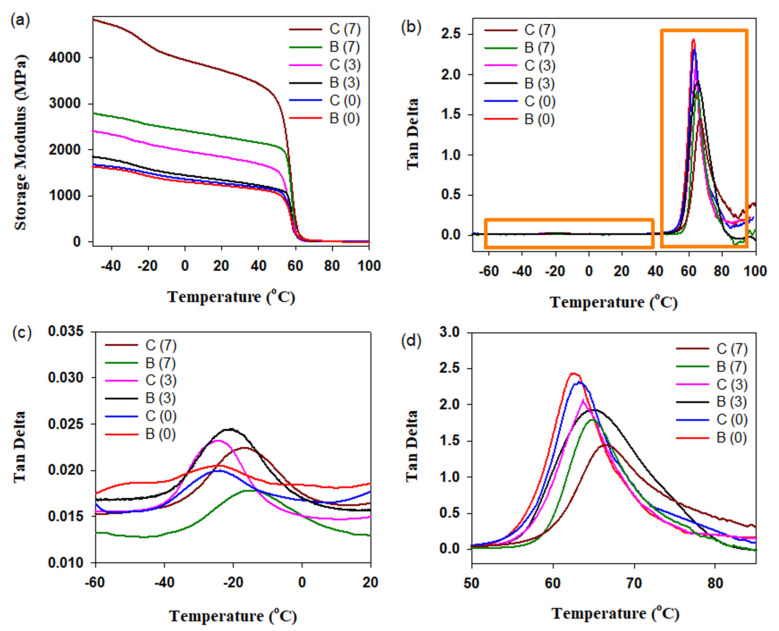
Storage modulus (**a**) and tan δ (**b**–**d**) of PLA/PBAT as well as PLA/PBAT/CNFs blends and in situ generated composites in a tensile mode. C and B stand for the composite and blend, respectively. The numbers in parentheses indicate the concentration of CNFs.

**Figure 4 polymers-13-01900-f004:**
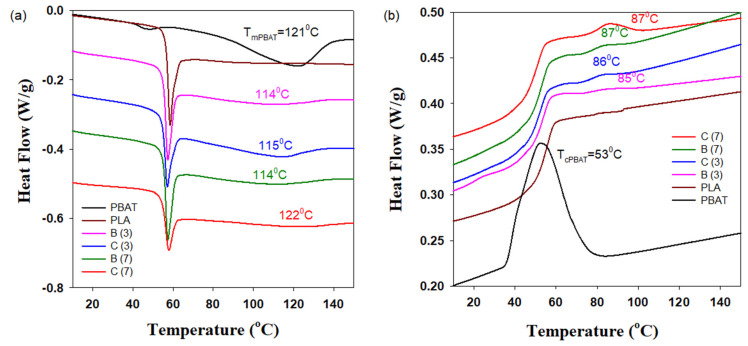
Heating endotherms (**a**) and cooling exotherms (**b**) of PLA and PBAT as well as PLA/PBAT/CNFs blends and in situ generated composites. C and B stand for the composite and blend, respectively. The numbers in parentheses indicate the concentration of CNFs.

**Figure 5 polymers-13-01900-f005:**
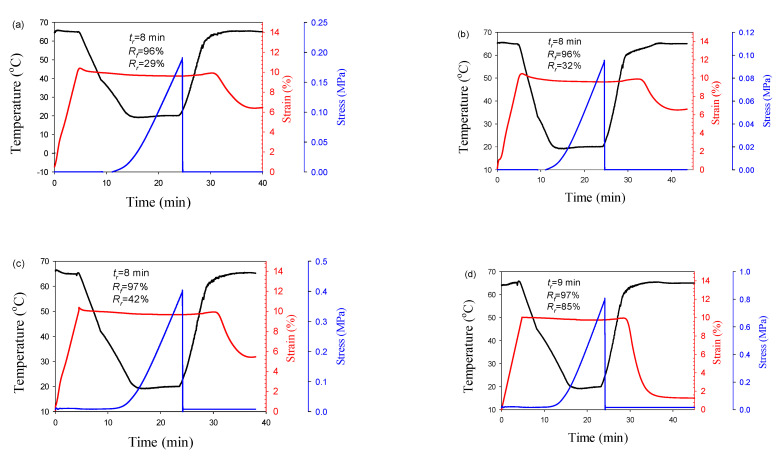
Temperature and strain of PLA/PBAT/CNFs blends (**a**,**b**) and in situ generated composites (**c**,**d**) during dual shape memory cycle. T_d_ = 65 °C. (**a**,**c**)–3%, (**b**,**d**)–7 wt.%CNTs.

**Figure 6 polymers-13-01900-f006:**
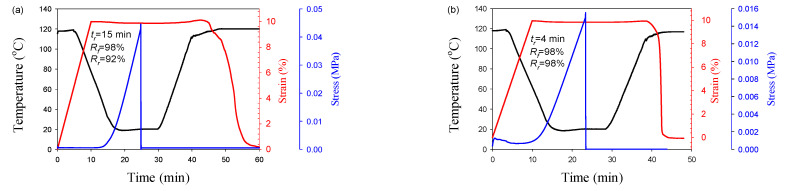
Temperature and strain of PLA/PBAT/CNFs blends (**a**) and in situ generated composites (**b**) during dual shape memory cycle. T_d_ = 120 °C. CNFs—7 wt.%.

**Figure 7 polymers-13-01900-f007:**
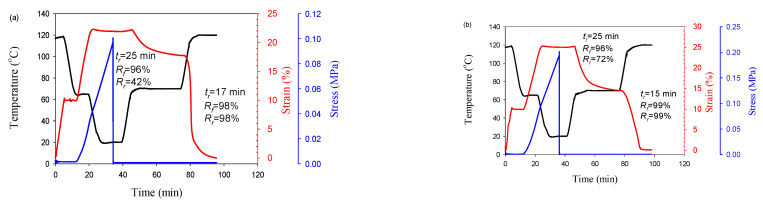
Temperature and strain of PLA/PBAT/CNFs blends (**a**) and in situ generated composites (**b**) during triple shape memory cycle. T_d_ = 65 °C, 120 °C. CNFs—7 wt.%.

**Table 1 polymers-13-01900-t001:** Mechanical properties of PLA/PBAT/CNFs blends and composites.

Material	CNFs, wt.%	Modulus of Elasticity (GPa)	Yield Stress (MPa)	Stress at Break (MPa)	Strain at Break (%)
Blend	0	1.64 ± 0.12	41.6 ± 1.4	51.2 ± 1.4	136.2 ± 11.1
	3	1.85 ± 0.09	42.2 ± 1.3	55.0 ± 0.6	12.1 ± 2.4
	7	2.28 ± 0.10	45.0 ± 1.6	56.7 ± 1.5	8.8 ± 2.0
Composite	0	2.35 ± 0.12	44.8 ± 2.1	66.7 ± 3.3	185.2 ± 9.0
	3	2.60 ± 0.13	47.2 ± 1.5	60.1 ± 3.3	24.5 ± 3.4
	7	2.92 ± 0.12	50.0 ± 1.6	62.7 ± 2.2	38.4 ± 2.9

Neat PLA: E = 2.04 + 0.09 GPa, σ_b_ = 43.0 ± 1.7 MPa, ε_b_ = 7.0 + 1.0%.

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
