# Peer review of "In Situ Generation of Green Hybrid Nanofibrillar Polymer-Polymer Composites—A Novel Approach to the Triple Shape Memory Polymer Formation"

_polymers, 2021, doi:10.3390/polym13121900_

Round 1

Reviewer 1 Report

In this study, authors have shown In-situ generation of nanofibrillar polymer-polymer composites for triple shape memory polymer formation. The manuscript had been carefully organized and could be accepted after below modification:

  1. The labels (a, b, c ,d…) in the image are too small to identify. Pls bold them and increase the font size.
  2. In Figure 3, the labels (a) to (b) are not uniform. Pls modify them.
  3. In many figures, there are lots of C(7), B(7) which should be descripted and discussed
  4. The format of Figures should be same in whole manuscript. Pls modify all those figures.
  5. The nanofibrillar should be proved by SEM. Authors should provide SEM image with higher resolution.

Author Response

  1. The labels (a, b, c ,d…) in the image are too small to identify. Pls bold them and increase the font size.

The appropriate font size of 12 was used for the labels in the edited manuscript. 

2. In Figure 3, the labels (a) to (b) are not uniform. Pls modify them.

The corresponding change was made to the manuscript.

3. In many figures, there are lots of C(7), B(7) which should be descripted and discussed

The corresponding change was made to the manuscript as below.

“In the whole text, C and B stand for the composite and blend, respectively. The numbers in parentheses indicate the concentration of CNFs.”

4. The format of Figures should be same in whole manuscript. Pls modify all those figures.

The figures were prepared using the Sigma plot software in a way to best represent the existing results. The final stage of figures will be fixed by the help of editorial board in the stage of publication. 

5.The nanofibrillar should be proved by SEM. Authors should provide SEM image with higher resolution.

The task of SEM studies was to show not only the size of the formed fibrils, but also the formation of a continuous network by them. If the reviewer does not object, we would leave the magnification we have chosen, which allows us to judge both the size of the fibrils and their spatial organization.

Reviewer 2 Report

The authors present a study where they have systematically investigated the shape memory properties of PLA/PBAT/cellulose after either blending or in situ fibrillation. They have found that the best shape memory properties could be achieved with 7% cellulose after fibrillization and that the polymer even exhibited two distinct phase transitions which allowed for triple shape programming/recovery with good fixation/recovery ratio.

I have some comments that should be addressed by the authors.

  1. 2 sample preparation: you may want to add more details for the fabrication of in-situ generated nanocomposites as is it not convenient for the reader to look up previous publications, especially if their institutions do not have full access to all publishers. Also a figure for the process could be helpful.
  2. 3 mechanical properties: more details would be valuable. E.g what was the specimen geometry for tensile tests and how were they cut? More details for DMA (clamping distance, pre-load, oscillation strain?)
  3. I only saw that it was mentioned how many samples were measured for the tensile tests, not for the other characterization methods. It would be important to have multiple measurements for each method to confirm the repeatability and reliability of the results reported. Please add or comment.
  4. For the shape memory properties. I was wondering what the heating/cooling rates were and what holding times at various temperatures were used. Also, what was the rationale for the deformations used (10 and 25%)? Please comment.
  5. What was the rationale for the deformation/recovery temperatures used? I do understand that you selected temperatures above the transition temperatures as determined by DSC. However, I would be better to use temperatures that are at least 10 degrees above the offset of these temperatures to archive better and faster transformations. You mentioned recovery at high temp took about 15 mins. That could change with a higher recovery temperature.
  6. Shape recovery: It is known that the shape recovery during the first cycle may differ drastically from adjacent cycles. I assume only one shape memory cycle was run per experiment? Please add this information. You may also want to consider running at least 5 cycles to get information on the repeatability and behavior for cycles after the first one. That will give you also values for Rr and R r, tot.
  7. In the introduction, it was mentioned that recovery forces for SMPs are usually low (lower as compared to SMAs). I was wondering if you have also performed shape memory experiments where you have measured the recovery forces?
  8. Figure 4: I would rename the caption to heating endotherms instead of melting since these curves show also glass transitions. You could also indicate the Tg(s) in the figure itself.
  9. Figs 5, 6, and 7: It would be informative if the stress would also be displayed.
  10. SME: You could consider showing a qualitative image of a specimen that undergoes the triple shape effect to nicely visualize that for the reader.
  11. Tensile tests: You have reported tensile tests for the different polymers investigated. I assume these were done at RT. (table 1). Here, one can see that elongation at break for the 7% B/C are about 9 and 38%. Yet, you stretch up to 25% in the second programming step. This will exceed the capacity of the blend. Therefore, it would be good if you could comment on this. Have you performed tensile tests at higher temperatures as well (the programming temperatures)? That would actually be good data to present here as well and may help to explain that behavior.

Author Response

I have some comments that should be addressed by the authors.

  1. 2 sample preparation: you may want to add more details for the fabrication of in-situ generated nanocomposites as is it not convenient for the reader to look up previous publications, especially if their institutions do not have full access to all publishers. Also a figure for the process could be helpful.

The subsequent change was added to the manuscript as below.

“In-situ generated nanocomposites were prepared by shearing of blends to allow formation of nanofibers and their shear induced crystallization. To avoid the degradation due to hydrolysis, PLA and PBAT were dried at 65 ˚C under vacuum before extrusion. Melt blending was carried out in a co-rotating twin screw extruder, 2x20/40D EHP Zamak Mercator, with a ratio of screw length to its diameter (L/D) of 40. Further blending was followed by extrusion of tapes using single screw extruder (PlastiCorder PLV 151, Brabender, L/D = 25) equipped with the 12 mm wide, 0.8 mm thick and 100 mm long slit die. The extrudates were cast on a transport belt at 25 °C. The extrudates in the form of tapes approximately 0.5 mm thick and 10 mm wide were obtained.”

  1. 3 mechanical properties: more details would be valuable. E.g what was the specimen geometry for tensile tests and how were they cut? More details for DMA (clamping distance, pre-load, oscillation strain?)

The subsequent change was added to the manuscript as below.

“Specimens of the gauge length of 25 mm and the width of 5 mm (ISO 527-2, type 1BA) were cut out from extruded tapes using a steel template.”

“The DMA studies were performed in a multi-frequency-strain mode using single cantilever fixture with a clamp size of 17.5 mm, strain set to 0.02 %.”

  1. I only saw that it was mentioned how many samples were measured for the tensile tests, not for the other characterization methods. It would be important to have multiple measurements for each method to confirm the repeatability and reliability of the results reported. Please add or comment.

The subsequent change was added to the manuscript as below.

“The tests for each material were repeated at least eight times, and the results were averaged.”

  1. For the shape memory properties. I was wondering what the heating/cooling rates were and what holding times at various temperatures were used. Also, what was the rationale for the deformations used (10 and 25%)? Please comment.

The method Log for dual-SME and triple-SME, containing the whole program set for SM studies, is added to the manuscript in the supplementary materials.

Method Log for dual-SME

Method Log for triple-SME

1: Force 0.003 N

2: Ramp 10.00°C/min to 65.00°C

3: Isothermal for 5.00 min

4: Data storage: On

5: Ramp strain 2.000 %/min to 10.000 %

6: Ramp 10.00°C/min to 20.00°C

7: Isothermal for 10.00 min

8: Force 0.003 N

9: Isothermal for 5.00 min

10: Ramp 10.00°C/min to 65.00°C

11: Isothermal for 45.00 min

12: Data storage: Off

13: End of method

1: Force 0.003 N

2: Ramp 10.00°C/min to 120.00°C

3: Isothermal for 2.00 min

4: Data storage: On

5: Ramp strain 2.000 %/min to 10.000 %

6: Ramp 10.00°C/min to 65.00°C

7: Isothermal for 2.00 min

8: Ramp strain 2.000 %/min to 25.000 %

9: Ramp 10.00°C/min to 20.00°C

10: Isothermal for 10.00 min

11: Force 0.003 N

12: Isothermal for 5.00 min

13: Ramp 10.00°C/min to 65.00°C

14: Isothermal for 30.00 min

15: Ramp 10.00°C/min to 120.00°C

16: Isothermal for 45.00 min

17: Data storage: Off

18: End of method

Due to the high plasticity of the matrix at elevated temperatures, the selected deformation values ere selected to avoid the yielding of samples at higher deformations. 

  1. What was the rationale for the deformation/recovery temperatures used? I do understand that you selected temperatures above the transition temperatures as determined by DSC. However, I would be better to use temperatures that are at least 10 degrees above the offset of these temperatures to archive better and faster transformations. You mentioned recovery at high temp took about 15 mins. That could change with a higher recovery temperature.

Previously we investigated the effect of temperature on the SM properties by performing the tests at 60, 65, and 70 ˚C. it was found that the best SM characteristics were achieved at 65 ˚C. Despite the fact that at higher temperatures the recovery takes place faster, on the other hand it will result in lower recovery ratios. Besides, for programming a triple shape memory effect, it was very important that the crystallization process of the PBAT should be completed before the glass-to-rubber transition of PLA. Otherwise, the programming of the second temporal shape will be accompanied by a partial recovery of the first temporal shape.

  1. Shape recovery: It is known that the shape recovery during the first cycle may differ drastically from adjacent cycles. I assume only one shape memory cycle was run per experiment? Please add this information. You may also want to consider running at least 5 cycles to get information on the repeatability and behavior for cycles after the first one. That will give you also values for Rr and R r, tot.

In this article, we wanted to show for the first time the possibility of creating non-covalent connections for the formation of reverse and permanent domains by in-situ transformation of polymer blend into polymer-polymer composites. The formed structure of in-situ generated polymer-polymer composite is quite stable ( it is even possible to reprocess it by injection molding, as was shown in one of our articles(R. Hosseinnezhad, I.Vozniak, J.Morawiec, A. Galeski, S. Dutkiewicz, In situ generation of sustainable PLA-based nanocomposites by shear induced crystallization of nanofibrillar inclusions. RSC Adv. 2015, 9, 30370). It can be assumed that shape recovery will not change much during cycling, but we would like to conduct more detailed research on the effect of cycling on shape recovery in our subsequent studies, if the reviewer does not object.

  1. In the introduction, it was mentioned that recovery forces for SMPs are usually low (lower as compared to SMAs). I was wondering if you have also performed shape memory experiments where you have measured the recovery forces?

The SP studies in our study was based on stress-strain measurement rather than force-displacement. Therefore, we did not have the absolute values to be included in the result.

  1. Figure 4: I would rename the caption to heating endotherms instead of melting since these curves show also glass transitions. You could also indicate the Tg(s) in the figure itself.

The caption  in Figure 4 is renamed to heating endotherms instead of melting.

  1. Figs 5, 6, and 7: It would be informative if the stress would also be displayed.

The stress values are added to the Figs 5, 6, and 7 in the edited manuscript.

  1. SME: You could consider showing a qualitative image of a specimen that undergoes the triple shape effect to nicely visualize that for the reader.

The SM studies were performed in DMA Q800 apparatus, where the sample was deformed using film tension fixtures. The chamber of DMA Q800 is not equipped with the monitoring system to capture the images during shape recovery. 

  1. Tensile tests: You have reported tensile tests for the different polymers investigated. I assume these were done at RT. (table 1). Here, one can see that elongation at break for the 7% B/C are about 9 and 38%. Yet, you stretch up to 25% in the second programming step. This will exceed the capacity of the blend. Therefore, it would be good if you could comment on this. Have you performed tensile tests at higher temperatures as well (the programming temperatures)? That would actually be good data to present here as well and may help to explain that behavior.

As it is mentioned in the manuscript, the tensile properties reported in table 1 were measured at room temperatures while the SM studies were performed at elevated temperatures making it possible to exceed the reported strain values for blend. Higher temperature of SM program alters the plastic deformation of the matrix resulting in higher ductility.